# Thiacalixarene Carboxylic Acid Derivatives as Inhibitors of Lysozyme Fibrillation

**DOI:** 10.3390/ijms25094721

**Published:** 2024-04-26

**Authors:** Anastasia Nazarova, Igor Shiabiev, Ksenia Shibaeva, Olga Mostovaya, Timur Mukhametzyanov, Arthur Khannanov, Vladimir Evtugyn, Pavel Zelenikhin, Xiangyang Shi, Mingwu Shen, Pavel Padnya, Ivan Stoikov

**Affiliations:** 1A. M. Butlerov Chemistry Institute, Kazan Federal University, 18 Kremlyovskaya Str., 420008 Kazan, Russia; 2Interdisciplinary Center of Analytical Microscopy, Kazan Federal University, 18 Kremlyovskaya Str., 420008 Kazan, Russia; 3Institute of Fundamental Medicine and Biology, Kazan Federal University, 18 Kremlyovskaya Str., 420008 Kazan, Russia; 4State Key Laboratory for Modification of Chemical Fibers and Polymer Materials, Shanghai Engineering Research Center of Nano-Biomaterials and Regenerative Medicine, College of Biological Science and Medical Engineering, Donghua University, Shanghai 201620, China; 5CQM—Centro de Química da Madeira, Universidade da Madeira, Campus Universitário da Penteada, 9020-105 Funchal, Portugal

**Keywords:** thiacalix[4]arene, lysozyme, HEWL, fibrillation, inhibition, cytotoxicity

## Abstract

Amyloid fibroproliferation leads to organ damage and is associated with a number of neurodegenerative diseases affecting populations worldwide. There are several ways to protect against fibril formation, including inhibition. A variety of organic compounds based on molecular recognition of amino acids within the protein have been proposed for the design of such inhibitors. However, the role of macrocyclic compounds, i.e., thiacalix[4]arenes, in inhibiting fibrillation is still almost unknown. In the present work, the use of water-soluble thiacalix[4]arene derivatives for the inhibition of hen egg-white lysozyme (HEWL) amyloid fibrillation is proposed for the first time. The binding of HEWL by the synthesized thiacalix[4]arenes (logKa = 5.05–5.13, 1:1 stoichiometry) leads to the formation of stable supramolecular systems capable of stabilizing the protein structure and protecting against fibrillation by 29–45%. The macrocycle conformation has little effect on protein binding strength, and the native HEWL secondary structure does not change via interaction. The synthesized compounds are non-toxic to the A549 cell line in the range of 0.5–250 µg/mL. The results obtained may be useful for further investigation of the anti-amyloidogenic role of thiacalix[4]arenes, and also open up future prospects for the creation of new ways to prevent neurodegenerative diseases.

## 1. Introduction

Amyloidosis is an uncommon group of diseases characterized by deposition of amyloids, i.e., protein aggregates with fibrillary morphology, in tissues and organs [1,2]. The formation of amyloid structures is caused by the misfolding of globular proteins and their subsequent aggregation into insoluble fibrils, which are characterized by a high content of β-sheets in their secondary structure. Nowadays, a number of neurodegenerative diseases, e.g., Alzheimer’s and Parkinson’s diseases and type 2 diabetes, have been shown to be associated with the formation of amyloid fibrils [3,4,5]. Amyloid deposition on the heart muscle is accompanied by its damage, leading to heart failure and arrhythmia [6,7,8]. Hen egg-white lysozyme (HEWL) has been widely used as a model protein for studying the processes of amyloid fibril formation due to its commercial availability and structural similarity to human lysozyme [9,10]. HEWL is a polypeptide chain consisting of 129 amino acid residues with four intramolecular disulfide bridges that account for its significant conformational stability.

One approach to control amyloidosis is to inhibit the formation of fibrillar aggregates [11,12]. Low-toxic organic compounds such as heterocyclic compounds [13,14,15], synthetic polymers [16,17], dendrimers [18], and low- and high-molecular-weight natural compounds [19,20,21] are generally used for this purpose. In recent years, macrocyclic compounds such as cyclodextrins, porphyrins, (thia)calixarenes, and pillararenes have been gaining popularity in biomedical applications [22,23,24,25,26,27,28]. This is easily explained by the great receptor properties of macrocycles with high-molecular-weight substrates, in particular proteins, due to multipoint supramolecular interactions of a large number of functional groups. Derivatives of (thia)calixarenes are among the most studied compounds capable of efficient protein binding [29,30,31,32,33]. At the same time, most publications are devoted to the complexation of proteins, e.g., cytochrome C [34,35], lysozyme [30], Penicillium antifungal protein [36], and bovine serum albumin [37], with calix[n]arenes (n = 4, 6, and 8) containing negatively charged sulfonate groups on the upper rim. There are also studies on protein complexation by (thia)calixarenes containing phosphonate, carboxylate, quaternary ammonium, amine, and PEG moieties [38,39,40,41,42,43,44,45,46]. However, there are only a small number of works on the effect of upper-rim substituted calixarene derivatives containing carboxylate and sulfonate groups on protein fibrillation [15,47,48,49]. At the same time, there are no examples in the literature that have studied the effect of thiacalixarenes substituted at the lower and/or upper rims and lower-rim-substituted calixarenes on protein fibril formation. Therefore, studies on the synthesis and anti-amyloidogenic activity of new derivatives of (thia)calixarenes are highly relevant.

It is well known that the binding of macrocyclic compounds to proteins can be due to hydrophobic interactions with the aromatic amino acid residues of the biomolecule or electrostatic interactions of oppositely charged components. In this regard, we hypothesized that the use of macrocyclic polyfunctional thiacalix[4]arenes would preserve the secondary structure of the protein via the interaction, thereby protecting the protein from damage and inhibiting HEWL fibril formation. In this work, a series of water-soluble thiacalix[4]arene octapropionic acid (**TOPA**) derivatives in different stereoisomeric forms (*cone*, *partial cone*, and *1,3-alternate*) were synthesized, and their cytotoxicity was evaluated. The interaction and aggregation of the obtained compounds with HEWL were studied by a series of physical methods, and their effect on fibril formation was shown.

## 2. Results and Discussion

### 2.1. Synthesis of TOPA Macrocycles

The aza-Michael reaction is a convenient way to obtain functionalized macrocyclic derivatives, as shown in our research [50,51,52]. A divergent approach has been proposed for the design of water-soluble negatively charged thiacalix[4]arenes. Previously obtained derivatives of *p-tert*-butylthiacalix[4]arene **1**–**3** in three main conformations (*cone*, *partial cone*, and *1,3-alternate*) [53] containing amide moieties and reactive primary amino groups at the lower rim were used as starting compounds. Thiacalixarenes **1**–**3** were introduced into the aza-Michael reaction with methyl acrylate in methanol according to the developed methods (Figure 1) [50]. Compounds **4**–**6** (*cone*, *partial cone*, and *1,3-alternate* conformations, respectively) containing eight ester moieties on the lower rim of the macrocycle were obtained in high yields.

Then, the hydrolysis of the ester fragments of compounds **4**–**6** was studied (Figure 1). The use of strong bases (e.g., sodium or potassium hydroxides) in this synthesis may lead to undesirable reactions since the structure of these compounds contains amide groups that are potentially also susceptible to hydrolysis. It is also difficult to isolate the target water-soluble compounds from aqueous solutions and remove excess sodium or potassium hydroxide. Our scientific team has previously shown [54,55] that the use of lithium hydroxide as a soft base led to the hydrolysis of ester fragments without the hydrolysis of amide and peptide bonds. Therefore, hydrolysis was carried out in a THF/H_2_O (*v*/*v* = 2:1) solvent mixture in the presence of lithium hydroxide at room temperature for 18 h. Intermediates in the form of iminodipropionic acid hydrochlorides on the thiacalix[4]arene platform were isolated after neutralization of the base with concentrated HCl. These intermediates were converted to the form of sodium salts by interaction with equimolar amounts of sodium hydroxide in a mixture of *i*PrOH:MeOH. Water-soluble thiacalix[4]arene octapropionic acid (**TOPA**) derivatives were obtained as sodium salts in *cone* (**TOPA-cone**), *partial cone* (**TOPA-paco**), and *1,3-alternate* (**TOPA-alt**) conformations (Figure 1).

The structure of **TOPA** macrocycles was confirmed by a series of physical methods, i.e., ^1^H, ^13^C NMR, FTIR spectroscopy, and ESI-HR mass spectrometry (Appendix A). Proton signals of ethylidene fragments were presented as two broadened triplets at 2.32–2.37 and 2.75–2.80 ppm in the ^1^H NMR (D_2_O) spectra of **TOPA** compounds (Appendix A). Methylene group signals of the hexylidene fragment at the tertiary nitrogen atom were presented as multiplets at 2.41–2.55 ppm. The protons described above were located at a sufficient distance from the thiacalix[4]arene platform and were almost unaffected by it. As a result, their chemical shifts for different macrocycle conformations were similar. The signals of *tert*-butyl, oxymethylene, and aromatic protons, as well as the methylene protons near the amide group, were characterized by different chemical shifts for each of the conformations. The signals of *tert*-butyl, oxymethylene, and aromatic protons appeared as singlets at 1.07, 4.89, and 7.46 ppm, respectively, in the case of **TOPA-cone** (Appendix A). The signals of the NHCH_2_ methylene group resonated as a multiplet at 3.24–3.28 ppm. In the case of **TOPA-alt** (Appendix A), the shielding of the above-mentioned protons by neighboring aromatic fragments was changed, which resulted in their resonating at 1.22, 4.50, 7.57 ppm (*tert*-butyl, oxymethylene, and aromatic fragments, respectively), and 2.97 ppm (methylene group near the amide fragment). The ^1^H NMR spectrum of **TOPA-paco** had more signals due to the lesser symmetry of its structure. Complete hydrolysis was confirmed by the absence of OCH_3_ methyl group signals in the ^1^H NMR spectra of all synthesized **TOPA** macrocycles.

**TOPA** macrocycles were also characterized by FTIR spectroscopy. Intense absorption bands at 1395 cm^−1^ and 1585 cm^−1^ characteristic of carboxylate compounds and corresponding to vibrations of COO^–^ fragments were observed in the FTIR spectra of **TOPA** (Appendix A). The second of the bands (1585 cm^−1^) also overlapped with the absorption band of amide II (1590 cm^−1^). Amide I and amide III bands were also observed at 1650 cm^–1^ and 1260 cm^–1^, respectively. An absorption band at 1095 cm^–1^ corresponding to arylalkyl ether vibrations (C_Ar_OCH_2_) was additionally present in the FTIR spectra of **TOPA**.

Intense signals of the negatively charged ion [M − 8Na + 6H]^2–^ corresponding to the charged form of the obtained carboxylate thiacalixarene derivatives without sodium ions were observed in the ESI-HR mass spectra of the synthesized compounds (Appendix A). The protonation is explained by the use of formic acid in classical techniques for mass spectrometry experiments. Nevertheless, the obtained data fully confirmed the structure of synthesized **TOPA** compounds.

Thus, a synthetic approach for the mild hydrolysis of thiacalix[4]arene ester derivatives and obtaining the targeted water-soluble carboxylate derivatives as sodium salts in three conformations (*cone*, *partial cone*, and *1,3-alternate*) has been developed and successfully implemented. The structure of the obtained compounds was confirmed by a series of physical methods.

### 2.2. Aggregation Properties of TOPA Macrocycles

The study of the self-association of macrocyclic molecules in water is one of the most interesting and promising directions in modern supramolecular chemistry [25,56,57,58,59]. Understanding and controlling the processes of supramolecular associates’ formation is of great importance for future applications of such aggregates, both in the field of biomedicine and in the creation of smart materials [60,61]. Therefore, the next stage of the work was the study of **TOPA** macrocycles’ self-association in water in the 1 µM–100 µM concentration range (Table 1) by dynamic light scattering (DLS). Thus, it was found (Appendix A) that the most stable particles are formed by **TOPA-cone** regardless of the solution concentration. Both the average hydrodynamic particle diameter (d) and the polydispersity index (PDI) of the systems decreased with increasing **TOPA-cone** concentration. Zeta potential (ζ) values of aggregates decreased (from −10.7 to −31.2 mV) with increasing concentrations (Appendix A). It is well known that the zeta potential value allows to assess the stability of colloidal systems. Previously, a zeta potential threshold of 30 mV modulo was experimentally established for stable systems [62]. Thus, **TOPA-cone** formed the most stable aggregates at a concentration of 100 µM, which is indicated by the lowest values of PDI (0.16) and ζ-potential (−31.2 mV). The formation of stable aggregates in the case of **TOPA-cone** is probably due to its structure, in which there is a clear separation of the molecule into polar (carboxylate functions) and non-polar parts (*tert*-butyl and aromatic fragments). This made **TOPA-cone** similar in its behavior to surfactant molecules, where stable aggregates are formed upon reaching some critical concentration. In turn, the symmetry violation of the macrocycle in the case of **TOPA-paco** led (Appendix A) to the formation of several fractions of submicron particles, while the system PDI increased (from 0.51 to 0.64) with increasing solution concentrations. **TOPA-alt** was characterized by the formation of nanometer and submicron aggregates (Appendix A), and PDI values of the systems varied unevenly with changes in concentration.

### 2.3. Cytotoxicity of TOPA Macrocycles

The study of the ability of the synthesized **TOPA** macrocycles to inhibit the viability and proliferative activity of the A549 cell line using an MTT assay was carried out according to the literature [63]. **TOPA** macrocycles had no ability to reduce A549 cell viability (Appendix A) over the 0.5–250 µg/mL (0.24–120 µM) concentration range.

### 2.4. Complexation of TOPA Macrocycles with HEWL

To date, the literature presents a huge number of studies on the inhibition of amyloid fibril formation [13,14,15,64,65]. One way to design such inhibitors is an approach based on the molecular recognition of amino acids in the native protein by the host molecule and their subsequent binding [48,66,67,68,69]. HEWL was chosen as a model protein due to its structural similarity to human lysozyme and commercial availability [9,10,70,71]. It is well known that a wide range of variously substituted (thia)calix[4]arenes are able to bind HEWL through both electrostatic interactions or π–π stacking with amino acid fragments [29,30,44,72,73]. In view of this, the next part of the work was to study the complexing properties of **TOPA** macrocycles with HEWL by UV-Vis spectroscopy, DLS, and transmission and scanning electron microscopy. Initially, the interaction of **TOPA** with HEWL was studied by UV-Vis spectroscopy. A characteristic feature of the electronic absorption spectra of thiacalixarene stereoisomers is the difference in the UV-Vis spectra of macrocycles in *1,3-alternate* conformation from others. Thus, a pronounced absorption maximum at 259 nm and a shoulder at 275 nm were observed (Figure 2) for **TOPA-alt**, while the absorption spectra were similar for thiacalixarenes in *cone* and *partial cone* conformations with a shoulder in the 275–300 nm region (Appendix A). HEWL and synthesized **TOPA** macrocycles had similar chromophore fragments. There was an overlap between the absorption spectra of protein and macrocycles caused by π–π* transitions of aromatic rings and n–π* transitions of carbonyl groups. Nevertheless, a hyperchromic effect was observed in the electronic absorption spectra (Figure 2 and Appendix A) regardless of the thiacalixarene conformation, which clearly indicates that the components of the mixtures interact with each other.

The isomolar series method (Job’s method) was used to establish the stoichiometry of complexation. The maximum was at a **TOPA**/HEWL ratio equal to 0.5 in the Job plots (Figure 2c and Appendix A), which corresponds to a complexation stoichiometry of 1:1. Association constants of the **TOPA**/HEWL system were determined by spectrophotometric titration (Appendix A). The data obtained were processed by the Bindfit application (Appendix A) at a 1:1 stoichiometry. The following values of log*K*_a_ were found: log*K*_a_ (**TOPA-cone**/HEWL) = 5.13, log*K*_a_ (**TOPA-paco**/HEWL) = 5.05, and log*K*_a_ (**TOPA-alt**/HEWL) = 5.09.

Analysis of the association constants of **TOPA**/HEWL systems showed that the macrocycle conformation had almost no effect on the protein binding strength. Protein binding can be caused both by hydrophobic interactions of the macrocyclic core with the hydrophobic part of the biomolecule, namely the *L*-Trp residues, as well as by electrostatic interactions of components with opposite charges (thiacalixarene is negatively charged, while HEWL is positively charged).

An important aspect of the interaction of host molecules with proteins is the preservation of the native structure of biomolecules, which is necessary for normal functioning [74,75]. In this regard, the next stage of the work was the study of the interaction of **TOPA** macrocycles with HEWL by circular dichroism (CD) spectroscopy, which made it possible to unambiguously assess the effect of **TOPA** macrocycles on the secondary structure of protein. The CD spectra were recorded at a macrocycle/protein ratio equal to 1:1 in 20 mM phosphate buffer (PBS) (pH 7.44) containing 20 mM NaCl. Thus, the CD spectrum of HEWL showed (Figure 3) a negative minimum at 207 nm and a shoulder at 226 nm, indicating a high content of α-helices in the secondary structure of the protein [76]. The addition of **TOPA** compounds to the HEWL solution does not lead to any visible changes in the CD spectra, which confirms the preservation of the native secondary structure of protein upon interaction with the studied thiacalix[4]arenes. Therefore, it can be concluded that the interaction of HEWL with **TOPA** compounds is primarily due to electrostatic interactions.

The resulting associates (**TOPA**/HEWL) were studied by DLS in deionized water, and the association of HEWL was also evaluated (Appendix A). HEWL formed a polydisperse system at a concentration equal to 10 µM with an average particle diameter of 3.1 ± 0.7 nm, which corresponds to its monomeric form. The macrocycle/HEWL ratio of 1:1 was chosen to study the aggregation of binary systems, since it was previously found that complexes of 1:1 stoichiometry (C**_TOPA_** = C_HEWL_ = 10 µM) were formed when protein interacted with the synthesized macrocycles. The study of the obtained systems by DLS showed that the formation of nano-sized associates occurred regardless of the macrocycle conformation (Table 2). Moreover, the formation of particles with the smallest mean hydrodynamic diameter (84 ± 2 nm) and the smallest PDI (0.11) was observed (Appendix A) only in the case of the **TOPA-paco**/HEWL mixture. There was a significant stabilization of the system in the presence of protein and a decrease in the size of the particles formed (from 298 ± 22 nm to 84 ± 2 nm) compared to the self-associates formed by **TOPA-paco** (Table 1). The addition of HEWL to the solutions of **TOPA-cone** and **TOPA-alt** resulted in particle enlargement (Appendix A). Moreover, the zeta potential values of the binary systems (Appendix A) were found to be close (Table 2) regardless of the macrocycle conformation. The isoelectric point of lysozyme is 11.1, which determines its positive charge in solution and interaction with **TOPA** due to electrostatic interactions. Eight carboxylate fragments are located on the same side of the macrocyclic platform in the case of **TOPA-cone**. Therefore, its interaction with HEWL may be most effective. The lipophilic *tert*-butyl fragments are located on the opposite side of the macrocyclic platform and interacted only with the same lipophilic *tert*-butyl groups of **TOPA-cone**, resulting in a layer-by-layer assembly of **TOPA-cone** with the protein. This assembly led to both the aggregates’ enlargement and the decreasing stability of such systems. The interaction of **TOPA-alt** with HEWL led to the most dramatic changes in particle diameter, as hypothesized above. In turn, six carboxyl moieties interacted with the HEWL molecule, and two carboxyl moieties were located on the opposite side of the macrocyclic platform in the case of **TOPA-paco**. It prevented layer-by-layer assembly compared to **TOPA-cone** and determined the smallest size of the formed aggregates.

Next, the resulting associates of **TOPA** macrocycles with HEWL were studied by transmission (TEM) and scanning electron microscopy (SEM) (Figure 4). Nanoparticles of various shapes were formed as a result of the association of **TOPA** with HEWL. Thus, submicron aggregates were formed (Figure 4a,d) for **TOPA-cone**/HEWL mixture (1:1). **TOPA-alt** with HEWL formed particles with a diameter of 400–500 nm, which stuck together and formed much larger supramolecular aggregates with an elongated structure in several growth directions (Figure 4c,f). Spherical aggregates with an average diameter of 100–200 nm were formed by **TOPA-paco** and HEWL. Their clumping together led to the formation of structures like bunches of grapes (Figure 4b,e). The difference in the sizes of nanoparticles obtained by DLS and TEM/SEM methods is apparently due to different sample preparation and research methods.

Thus, the macrocycle conformation was found to have little effect on protein binding strength, while the native secondary structure of HEWL did not change under interaction with the synthesized thiacalix[4]arenes. The most stable binary system was formed for **TOPA-paco** according to DLS data, which was due to the absence of layer-by-layer protein–macrocycle–protein assembly and determined the smallest size of the formed aggregates.

### 2.5. The Inhibitory Effects of TOPA on HEWL Fibrillation

Inhibition of fibril formation is necessary to create new approaches in the prevention of neurodegenerative diseases. Therefore, the final stage of the work was to study the effect of **TOPA** macrocycles on the ability to inhibit HEWL fibril formation. Various fibril growing protocols have been described in the literature [14,77,78]. The approach proposed by Dasgupta’s group [79] was chosen, where the minimum required amount of ethanol was established for the formation of HEWL amyloid fibrils. Thus, HEWL fibrillation was performed in 20 mM PBS (pH = 7.44) with the addition of 30% (by volume) ethanol. HEWL was incubated at a concentration of 150 µM in the absence or presence of different concentrations of **TOPA** macrocycles (75, 150, and 300 µM). The solutions were incubated for 6 h at 65 °C with further incubation for 24 h at room temperature.

One of the widely used dyes for fibril detection is thioflavin T (ThT). The fluorescence quantum yield of ThT is significantly increased under incorporation into fibrils [80]. This phenomenon is due to its ability to bind to aromatic amino acid residues located in a specific way in β-sheets, which mainly represent the secondary structure of HEWL fibrils. No conformation influence on the fluorescence spectra of **TOPA** macrocycles with ThT was found (Appendix A). A decrease in ThT fluorescence was found when **TOPA** macrocycles were added to HEWL solutions during incubation (Figure 5). The intensity of ThT fluorescence decreased with increasing concentrations of thiacalix[4]arenes.

A uniform decrease in ThT fluorescence was observed with an increasing macrocycle/HEWL ratio (Figure 5a) for **TOPA-cone**. A decrease in ThT fluorescence was close to that at macrocycle/protein ratios of 1:2 and 1:1 in the case of **TOPA-paco** (Figure 5b). An addition of 75 μM of **TOPA-alt** did not change ThT fluorescence, while further increasing the macrocycle concentration resulted in a decrease in ThT fluorescence (Figure 5c). The data obtained clearly indicated the inhibition of HEWL fibril formation in the presence of the synthesized thiacalix[4]arenes. The effect of macrocycle conformation on the inhibition of fibril formation was found to be in good agreement with the data obtained by DLS for the macrocycle–protein systems (Table 2). A decrease in ThT fluorescence by 45% was observed for **TOPA-cone** at the 300 μM macrocycle concentration compared to the fluorescence spectrum of ThT with HEWL fibrils (Figure 5a). There was a decrease in ThT fluorescence for **TOPA-paco** (29%) and **TOPA-alt** (31%). This is probably due to the fact that **TOPA-cone** bound native HEWL by eight carboxylate fragments located on one side of the macrocyclic platform, making its interaction with HEWL the most effective. It has previously been shown that an addition of a 1.5-fold excess of 3,4,5-trihydroxybenzoic acid led to a decrease in the fluorescence intensity of ThT by ~68% [79], and incubation of HEWL in the presence of catechol reduced ThT fluorescence intensity by ~69% [77], while a 15-fold excess of ascorbic acid decreased the fluorescence intensity of the dye by almost 97% [78]. In the above examples, the inhibition of fibril formation occurs by a mechanism different from the mechanism associated with the molecular recognition of individual protein components by the host molecule. The examples describing the effect of macrocycles on the inhibition of HEWL fibril formation are very limited. However, J. Mohanty’s group [66] showed inhibition of HEWL fibril formation in the presence of sulfobutylether-β-cyclodextrin, which led to a decrease in the fluorescence intensity of ThT by almost three times. Therefore, it may be concluded that the results obtained in this work (considering the low macrocycle/lysozyme ratio) are promising in comparison with the literature examples.

One of the ways to assess the secondary structure of the amyloid fibrils formed is to analyze the FTIR spectra of the samples, i.e., the evaluation of the amide I, amide II, and amide III spectral bands [81,82]. It is well known that the amide I band in FTIR spectra allows to estimate the contribution of β-sheet content (1635–1610 cm^–1^), α-helices (1660–1650 cm^–1^), antiparallel β-sheets, and β-turns (1695–1665 cm^–1^). The conformation of HEWL after incubation with/without **TOPA** macrocycles was further studied by FTIR spectroscopy. A peak with the highest intensity at 1625 cm^–1^ (characteristic of β-sheets) was observed in the FTIR spectrum (Figure 6) of HEWL fibrils. Deconvolution of the FT-IR spectra in the target region was carried out. Changes in the ratio of integral areas are shown in Figure 6. The trend in the peak areas of the α-helix and β-sheet confirms the previously made assumption about thiacalix[4]arene’s role in the stabilization of the protein by binding native HEWL and thereby preventing its fibrillation (Appendix A). All TOPA macrocycles did not have signals in the region responsible for β-sheet signals according to the data obtained by FT-IR spectroscopy. However, the α-helix signals are extremely close to the signals of the amide groups of macrocycles. Therefore, for this experiment, all spectra of macrocycles with HEWL fibrils were retaken (Appendix A) using the corresponding macrocycle as a baseline, in the range from 1450 to 1800 cm^–1^. Thanks to this and the deconvolution carried out, the influence of macrocycles was shown more clearly and assessed more effectively. A decrease in the intensity of the band at 1625 cm^−1^ after incubation of HEWL with **TOPA** (macrocycle/protein ratio = 1:1) was shown, indicating a decrease in the number of β-sheets in the structure of amyloid fibrils. The greatest decrease in the band intensity at 1625 cm^–1^ was detected for **TOPA-cone** (Figure 6C), which is in good agreement with the previously obtained data on the higher binding efficiency of native HEWL by **TOPA-cone**. However, the band at 1655 cm^–1^ indicating the presence of α-helices in the secondary structure of the fibrils remained unchanged in the FTIR spectra (Figure 6) regardless of the presence/absence of **TOPA** during HEWL incubation. This fact indicated that studied **TOPA** macrocycles do not have the ability to disaggregate fibrils. Nevertheless, the obtained results confirmed that the synthesized thiacalix[4]arenes stabilized the protein by binding native HEWL and, thereby, prevented its fibrillation.

The formation of HEWL amyloid fibrils with/without **TOPA** macrocycles was also studied by TEM. The morphology of HEWL amyloid fibrils after incubation with/without **TOPA** macrocycles is shown in Figure 7. The maximum fibril formation was observed (Figure 7a) with a larger size and denser packing without **TOPA** compounds. The addition of **TOPA** to HEWL during incubation resulted in a significant decrease (Figure 7b–d) in the amount of amyloid fibrils in the samples. TEM images (Figure 7b–d) showed dendritic structures with small inclusions of short-length filamentous aggregates, characteristic of the fibrillary form of the protein. These results confirmed the observations obtained by fluorescence and FTIR spectroscopy about the prevention of HEWL fibril formation during incubation of the protein with macrocyclic **TOPA** derivatives.

The results of this study demonstrated the high efficiency of **TOPA** macrocycles in inhibiting HEWL fibril formation. This was explained by the electrostatic interaction between the negatively charged carboxylate fragments of thiacalixarenes and the positively charged protein. This interaction led to the formation of stable supramolecular macrocycle/HEWL systems capable of stabilizing the protein structure and protecting against fibrillation.

## 3. Materials and Methods

### 3.1. General Experimental Information

Detailed information on the equipment, methods, and physical–chemical characterization is presented in the Appendix A.

### 3.2. Synthesis

Thiacalix[4]arene derivatives **1**–**3** and **4**–**6** (in *cone*, *partial cone,* and *1,3-alternate* conformations, respectively) were synthesized according to previously described procedures [50,53].

#### General Procedure for the Synthesis of **TOPA**

The lithium hydroxide monohydrate (1.49 g, 35.6 mmol) was added to the solution of **4**–**6** (1.51 g, 0.742 mmol) in 60 mL of THF:H_2_O mixture (2:1). The reaction mixture was stirred for 18 h at room temperature. Afterwards, the excess of base was quenched by slow addition of 4 mL of HCl (conc.). Then, THF was removed on a rotary evaporator, and the reaction mixture was cooled to 0 °C. Thereafter, the precipitate formed was separated from the remaining solution by decantation and dried under reduced pressure.

The dry precipitate was dissolved in 25 mL of *i*PrOH:MeOH (5:1) mixture and the 12.4-fold molar excess of NaOH solution in 4 mL of methanol was added to this mixture and stirred for 10 min. The reaction mixture was then cooled to 5 °C. Afterwards, the resulting precipitate was separated by centrifugation, washed with cold *i*PrOH (15 mL), and dried under reduced pressure over phosphorus pentoxide.

**5,11,17,23-Tetra-*tert*-butyl-25,26,27,28-tetrakis[*N*-(6-(*N*,*N*-di(carboxyethyl)amino)hexyl)carbamoylmethoxy]-2,8,14,20-tetrathiacalix[4]arene octasodium salt (TOPA-cone)** in *cone* conformation. Yield: 1.28 g (82%). White powder, mp 245 °C.

^1^H NMR (D_2_O, δ, ppm): 1.07 (s, 36H, (CH_3_)_3_C), 1.20–1.30 (m, 16H, C(O)NHCH_2_CH_2_CH_2_CH_2_, C(O)NHCH_2_CH_2_CH_2_CH_2_), 1.39–1.57 (m, 16H, CH_2_CH_2_N, C(O)NHCH_2_CH_2_CH_2_CH_2_), 2.32 (br.t, 16H, NCH_2_CH_2_C(O)O^−^), 2.42–2.50 (m, 8H, CH_2_CH_2_N), 2.75 (br.t, 16H, NCH_2_CH_2_C(O)O^−^), 3.24–3.28 (m, 8H, C(O)NHCH_2_), 4.89 (s, 8H, OCH_2_C(O)), 7.46 (s, 8H, Ar-H).

^13^C{^1^H} NMR (D_2_O, δ, ppm): 25.42, 26.42, 26.84, 29.14, 31.09, 33.70, 34.06, 39.48, 49.72, 52.99, 74.19, 128.48, 134.86, 147.22, 157.93, 169.58, 180.75.

FTIR (ν, cm^−1^): 3334 (N-H), 1657 (C(O)NH, amide I), 1588 (C(O)NH, amide II; COO^−^), 1394 (COO^−^), 1266 (C(O)NH, amide III), 1095 (C_Ar_OCH_2_).

ESI HRMS: calculated [M − 8Na + 6H]^2–^ = 959.9532. Found [M − 8Na + 6H]^2–^ = 959.9524.

**5,11,17,23-Tetra-*tert*-butyl-25,26,27,28-tetrakis[*N*-(6-(*N*,*N*-di(carboxyethyl)amino)hexyl)carbamoylmethoxy]-2,8,14,20-tetrathiacalix[4]arene octasodium salt** (**TOPA-paco)** in *partial cone* conformation. Yield: 1.20 g (77%). White powder, mp 218 °C.

^1^H NMR (D_2_O, δ, ppm): 1.03 (s, 18H, (CH_3_)_3_C), 1.18–1.60 (m, 24H, C(O)NHCH_2_CH_2_CH_2_CH_2_, C(O)NHCH_2_CH_2_CH_2_CH_2_, CH_2_CH_2_N), 1.27 (s, 9H, (CH_3_)_3_C), 1.32 (s, 9H, (CH_3_)_3_C), 1.44–1.61 (m, 8H, C(O)NHCH_2_CH_2_CH_2_CH_2_), 2.37 (br.t, 16H, NCH_2_CH_2_C(O)O^−^), 2.41–2.55 (m, 8H, CH_2_CH_2_N), 2.80 (br.t, 16H, NCH_2_CH_2_C(O)O^−^), 3.15–3.38 (m, 8H, C(O)NHCH_2_), 4.58 (s, 2H, OCH_2_C(O)), 4.63–4.79 (m, 6H, OCH_2_C(O)), 7.23 (s, 2H, Ar-H), 7.55 (s, 2H, Ar-H), 7.80 (s, 2H, Ar-H), 7.84 (s, 2H, Ar-H).

^13^C{^1^H} NMR (D_2_O, δ, ppm): 25.30, 25.99, 26.07, 26.29, 26.35, 26.68, 28.82, 30.61, 30.77, 33.65, 33.72, 33.90, 33.99, 39.13, 39.51, 49.70, 52.78, 71.27, 72.24, 73.46, 125.88, 126.48, 127.87, 133.35, 134.89, 135.27, 135.82, 147.44, 148.06, 156.50, 157.32, 158.46, 169.92, 170.29, 170.39, 180.83.

FTIR (ν, cm^−1^): 3301 (N-H), 1662 (C(O)NH, amide I), 1590 (C(O)NH, amide II; COO^−^), 1382 (COO^−^), 1265 (C(O)NH, amide III), 1089 (C_Ar_OCH_2_).

ESI HRMS: calculated [M − 8Na + 6H]^2–^ = 959.9532. Found [M − 8Na + 6H]^2–^ = 959.9502.

**5,11,17,23-Tetra-*tert*-butyl-25,26,27,28-tetrakis[*N*-(6-(*N*,*N*-di(carboxyethyl)amino)hexyl)carbamoylmethoxy]-2,8,14,20-tetrathiacalix[4]arene octasodium salt** (**TOPA-alt)** in *1,3-alternate* conformation. Yield: 1.41 g (91%). White powder, mp 260 °C.

^1^H NMR (D_2_O, δ, ppm): 1.22 (s, 36H, (CH_3_)_3_C), 1.15–1.26 (m, 16H, C(O)NHCH_2_CH_2_CH_2_CH_2_, C(O)NHCH_2_CH_2_CH_2_CH_2_), 1.31–1.40 (m, 8H, CH_2_CH_2_N), 1.41–1.50 (m, 8H, C(O)NHCH_2_CH_2_CH_2_CH_2_), 2.35 (br.t, 16H, NCH_2_CH_2_C(O)O^−^), 2.45–2.55 (m, 8H, CH_2_CH_2_N), 2.79 (br.t, 16H, NCH_2_CH_2_C(O)O^−^), 2.95–2.99 (m, 8H, C(O)NHCH_2_), 4.50 (s, 8H, OCH_2_C(O)), 7.57 (s, 8H, Ar-H).

^13^C{^1^H} NMR (D_2_O, δ, ppm): 25.17, 26.16, 26.55, 29.08, 30.48, 33.46, 34.13, 39.96, 49.61, 52.63, 69.02, 126.99, 129.87, 148.87, 155.11, 169.61, 180.69.

FTIR (ν, cm^−1^): 3313 (N-H), 1659 (C(O)NH, amide I), 1570 (C(O)NH, amide II; COO^−^), 1395 (COO^−^), 1265 (C(O)NH, amide III), 1087 (C_Ar_OCH_2_).

ESI HRMS: calculated [M − 8Na + 6H]^2–^ = 959.9532. Found [M − 8Na + 6H]^2–^ = 959.9515.

## 4. Conclusions

A successful synthetic approach to mild hydrolysis of thiacalix[4]arene ester derivatives and obtaining the corresponding water-soluble carboxylate derivatives as sodium salts in three conformations (*cone*, *partial cone*, and *1,3-alternate*) was proposed. The macrocycle in *cone* conformation formed the most stable nano-sized self-associates, with average hydrodynamic diameters of 30–72 nm. The synthesized compounds were non-toxic against the A549 cell line. The binding ability of the synthesized thiacalix[4]arenes with HEWL was studied by UV-Vis, CD spectroscopy, DLS, and electron microscopy (TEM and SEM). It was found that the logarithm of association constants for all synthesized thiacalixarenes with HEWL ranged from 5.05 to 5.13 (1:1 stoichiometry), while the native secondary structure of HEWL did not change under interaction with the synthesized macrocycles. The most stable binary system was formed for the thiacalix[4]arene in *partial cone* conformation according to the DLS data (d = 84 ± 2 nm, PDI = 0.11), which was due to the absence of layer-by-layer protein–macrocycle–protein assembly and determined the smallest size of the formed aggregates. The thiacalix[4]arene in *cone* conformation was a more effective inhibitor of HEWL fibril formation (decrease in ThT fluorescence intensity by 45%), which was explained by the electrostatic interaction of negatively charged carboxylate fragments of thiacalix[4]arenes with positively charged protein. This interaction led to the formation of stable supramolecular macrocycle/HEWL systems capable of stabilizing the protein structure and protecting it from fibrillation by 29–45%. Hopefully, the obtained results will be useful for further investigation of the anti-amyloidogenic role of macrocyclic compounds, for creating new ways to prevent neurodegenerative diseases.

## Data Availability

The data presented in this study are available in the Appendix A.

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
