# Peer review of "Thiacalixarene Carboxylic Acid Derivatives as Inhibitors of Lysozyme Fibrillation"

_ijms, 2024, doi:10.3390/ijms25094721_

Round 1

Reviewer 1 Report

Comments and Suggestions for Authors

I do not recommend this paper for acceptance. If so, until the major revision. 

1. In the introduction, explain the first sentence, fibrillar glycoprotein amyloid...? Where did you get that glycoprotein from? Does it have to be a glycoprotein?

2. In line 50: How is HEWL similar to amyloid-β -protein when one is globular and the other IDP?

3. Figure 6: In FTIR, I do not see significant changes in the structure. Perform FTIR deconvolution and add a table to the article. How did you know how to measure the FTIR of fibrils with TOPA, when TOPA itself has a significant FTIR spectrum in that area?

4. Figure 7: I don't see fibrils, only beautiful salt. Make new images, with better resolution. Have you done the TEM correctly? Don't you need to do it with uranyl acetate and carbonation? Make a new TEM where the fibrils will also be visible, you have too large a scale of 2 µm. 

5. Complete the fluorescence spectra of TOPA with ThT. Complete all FTIR controls. 

Author Response

Reviewer #1

First of all, we would like to thank respected Reviewer for careful consideration of the manuscript. In accordance with the comments, the following changes have been made (please see Figures in the attached file):

  1. In the introduction, explain the first sentence, fibrillar glycoprotein amyloid...? Where did you get that glycoprotein from? Does it have to be a glycoprotein?

We are very grateful for your remark. There was an unfortunate typo.

The previous sentence “Amyloidosis is an uncommon group of diseases characterized by deposition of the fibrillar glycoprotein amyloid in tissues and organs” has been changed to “Amyloidosis is an uncommon group of diseases characterized by deposition of amyloids, i.e., protein aggregates with fibrillary morphology, in tissues and organs” and highlighted in green.

  1. In line 50: How is HEWL similar to amyloid-β -protein when one is globular and the other IDP?

Thank you for your remark, it was a serious misunderstanding. The sentence “In addition, amyloid fibrillation of HEWL is similar to amyloid-β protein fibrillation associated with Alzheimer's disease” has been deleted.

  1. 3. Figure 6: In FTIR, I do not see significant changes in the structure. Perform FTIR deconvolution and add a table to the article. How did you know how to measure the FTIR of fibrils with TOPA, when TOPA itself has a significant FTIR spectrum in that area?

Deconvolution of the FTIR spectra in the target region was carried out. Changes in the ratio of integral areas are shown in Figure 6. The trend in the peak area of α-helix and β-sheet confirms the previously made assumption about thiacalix[4]arene role in stabilization of the protein by binding native HEWL and thereby prevented its fibrillation. All TOPA macrocycles did not have signals in the region responsible for β-sheet signals according to data obtained by FTIR spectroscopy. However, the α-helix signals are extremely close to the signals of the amide groups of macrocycles. Therefore, for this experiment, all spectra of macrocycles with HEWL fibrils were retaken (Fig. S43) using the corresponding macrocycle as a baseline, in the range from 1450 to 1800 cm–1. Thanks to this and the deconvolution performed, the influence of macrocycles was shown more clearly and assessed more effectively. Detailed fit parameters are presented in Supplementary information Figs. S39-S42.

Figure 6. FTIR spectra of HEWL fibrils with/without TOPA macrocycles with deconvolution and indicating the ratios of the integral areas of α-helices to β-sheets: A) HEWL fibrils; B) HEWL fibrils + TOPA-paco; C) HEWL fibrils + TOPA-cone; D) HEWL fibrils + TOPA-alt.

  1. Figure 7: I don't see fibrils, only beautiful salt. Make new images, with better resolution. Have you done the TEM correctly? Don't you need to do it with uranyl acetate and carbonation? Make a new TEM where the fibrils will also be visible, you have too large a scale of 2 µm.

TEM images of HEWL amyloid fibrils with/without TOPA macrocycles were retaken (Fig. 7). Addition of TOPA to HEWL during incubation resulted in a significant decrease (Figs. 7b–7d) in the amount of amyloid fibrils in the samples. These results confirmed the observations obtained by fluorescence and FTIR spectroscopy about prevention of HEWL fibril formation during incubation of the protein with macrocyclic TOPA derivatives.

Figure 7. TEM images of HEWL amyloid fibrils formed with/without TOPA after incubation at pH 7.44 at 30% ethanol heated at 65 °C for 6 h and further incubation at room temperature for 1 day: a) without TOPA; b) with TOPA-cone; c) with TOPA-paco; d) with TOPA-alt.

  1. Complete the fluorescence spectra of TOPA with ThT. Complete all FTIR controls.

The interaction of macrocycles TOPA with thioflavin T (ThT) was studied in 20 mM PBS (pH = 7.44). No conformation influence on the fluorescence spectra of TOPA-cone + ThT (Fig. S38a), TOPA-paco + ThT (Fig. S38b), and TOPA-alt + ThT (Fig. S38c) mixtures was found. Figure below was also added to in Supplementary information Fig. S38.

Figure S38. Fluorescence spectra of: a) TOPA-cone with ThT; b) TOPA-paco with ThT; c) TOPA-alt with ThT in 20 mM PBS (Cmacrocycle = CThT = 10 µÐœ).

All TOPA macrocycles did not have signals in the region responsible for β-sheet signals according to data obtained by FTIR spectroscopy. However, the α-helix signals are extremely close to the signals of the amide groups of macrocycles. Therefore, for this experiment, all spectra of macrocycles with HEWL fibrils were retaken using the corresponding macrocycle as a baseline, in the range from 1450 to 1800 cm–1. Thanks to this and the deconvolution carried out, the influence of macrocycles was shown more clearly and assessed more effectively. We are really grateful the reviewer for this methodological comment.

Reviewer 2 Report

Comments and Suggestions for Authors

The authors synthesized and characterized a group of thiacalixarene carboxylic acid derivatives and subsequently tested them as the potential inhibitors of lysozyme fibrillation. It is no easy task to design and synthesize those compounds. The authors tried to demonstrate that the in-house TOPA macrocycles could prevent the fibrillation of HWEL protein with low cytotoxicity. Presented data are comprehensive, especially for the structure characterization. Hopefully, there will be follow-up studies to evaluate the promising functions of these compounds in vitro and in vivo. Below please find my comments for consideration.

1.   Figure 2, Consider using a same scale on the y-axis in panel a and b. Present formatting makes it difficult to compare axis between absorbance and wavelength. Panel a: Data regarding TOPA-cone and TOPA-paco are missing.

2. Figure 2. Modify the arrow. Specify the meaning/significance of the arrow.

3. Figure 4. Include Figures S-38&39 as two additional panel here.

4. In supporting figures regarding particle size analysis. Use D10, D50, D90 to summarize particle size distribution. Additionally, label/name colored distribution curves. What do different colors refer to? Clarify why some figures have multiple curves (e.g., Fig S13) while others have only one (e.g., Fig S22).

5. Experimental details regarding sections 2.4 & 2.5 are missing in Section 3 . This issue ought to be addressed. Some text in sections 2.4 & 2.5 should be moved to Materials and Methods.

6. Revise the Conclusions section to avoid repetition of the abstract's content and provide additional insights.

7. No need to use bold to highlight "TOPA", "TOPA-cone", "TOPA-alt", "TOPA-paco" throughout the manuscript

8. Figure S25. Consider performing statistic analysis (e.g., Lin's CCC) to compare the data.

9. Figure S26. Modify the arrow.

10. Lines 31-32 and Lines 294-296. The authors claim that "The macrocycle conformation was found to have little effect on protein binding strength, while the native secondary structure of HEWL did not change under interaction with the synthesized thiacalix[4]arenes". But the data in Table 2 Figure 4, and Figures S38-39 suggest something very much otherwise. While the authors' explanation in lines 288-290 is confusing. How come samples were not prepared and analyzed in a consistent manner? Experimental details should have been included. Also see comment#5.

11. Supporting figures. Figure captions should be place below the corresponding figure. 

12. Too many references (in total 85). Some cited papers are from the same authors, verging on self-citation.

Author Response

Reviewer #2

First of all, we would like to thank respected Reviewer for careful consideration of the manuscript. In accordance with the comments, the following changes have been made:

  1. Figure 2, Consider using a same scale on the y-axis in panel a and b. Present formatting makes it difficult to compare axis between absorbance and wavelength. Panel a: Data regarding TOPA-cone and TOPA-paco are missing.

Thank you for your recommendation. Y-axis have been changed in Figure 2.

You can find UV-Vis spectra of TOPA-cone and TOPA-paco in Supplementary Information, Figure S26.

  1. Figure 2. Modify the arrow. Specify the meaning/significance of the arrow.

Arrows have been modified in Figures 2 and S26, the necessary changes have been made.

  1. Figure 4. Include Figures S-38&39 as two additional panel here.

Figure 4 has been changed.

  1. In supporting figures regarding particle size analysis. Use D10, D50, D90 to summarize particle size distribution. Additionally, label/name colored distribution curves. What do different colors refer to? Clarify why some figures have multiple curves (e.g., Fig S13) while others have only one (e.g., Fig S22).

DLS images with size distribution of particles by number have been rechanged and added to Supplementary information. The d10, d50, and d90 values were added to DLS images. Figures with zeta potential distribution (S22-S24, S35-S37) represent the mean values of zeta potentials. In view of this only one curve has been shown on figures with zeta potential distribution. Figures with zeta potential distribution of the particles were checked and remade.

  1. Experimental details regarding sections 2.4 & 2.5 are missing in Section 3. This issue ought to be addressed. Some text in sections 2.4 & 2.5 should be moved to Materials and Methods.

We fully agree with this statement. You can find all experimental details regarding Sections 2.4 & 2.5 in Supplementary Information, namely Sections 1.5-1.9 in Supplementary section.

  1. Revise the Conclusions section to avoid repetition of the abstract's content and provide additional insights.

Conclusions has been rewritten. The following text has been added to the manuscript and highlighted in green:

“Successful synthetic approach to mild hydrolyze of thiacalix[4]arene ester derivatives and obtaining the corresponding water-soluble carboxylate derivatives as sodium salts in three conformations (cone, partial cone, and 1,3-alternate) was proposed. The macrocycle in cone conformation were formed the most stable nano-sized self-associates with average hydrodynamic diameters 30–72 nm. The synthesized compounds were non-toxic against A549 cell line. Binding ability of the synthesized thiacalix[4]arenes with HEWL was studied by UV-Vis, CD spectroscopy, DLS, and electron microscopy (TEM and SEM). It was found that the logarithm of association constants for all synthesized thiacalixarenes with HEWL ranged from 5.05 to 5.13 (1:1 stoichiometry), while the native secondary structure of HEWL did not change under interaction with the synthesized macrocycles. The most stable binary system was formed for the thiacalix[4]arene in partial cone conformation according to the DLS data (d = 84 ± 2 nm, PDI = 0.11), which was due to the absence of layer-by-layer protein-macrocycle-protein assembly and determined the smallest size of the formed aggregates. The thiacalix[4]arene in cone conformation was a more effective inhibitor of HEWL fibril formation (decrease of ThT fluorescence intensity by 45%) which was explained by the electrostatic interaction of negatively charged carboxylate fragments of thiacalix[4]arenes with positively charged protein. This interaction led to the formation of stable supramolecular macrocycle / HEWL systems capable of stabilizing the protein structure and protecting from fibrillation by 29–45%. Hopefully, the obtained results will be useful for further investigation of the anti-amyloidogenic role of macrocyclic compounds for creating new ways to prevent neurodegenerative diseases.”

  1. No need to use bold to highlight "TOPA", "TOPA-cone", "TOPA-alt", "TOPA-paco" throughout the manuscript

We fully agree with this statement. It is not necessary, but it is so good-looking and makes them convenient and visible to the readers.

  1. Figure S25. Consider performing statistic analysis (e.g., Lin's CCC) to compare the data.

Cytotoxicity of TOPA macrocycles in 0.5-250 μg/mL concentration range showed no significant differences compared with the no treatment option.

The following sentence has been added to Supplementary information:

“Three series of experiments were carried out with at least 8 replications for each variant in the series. For MTT test data significant differences were reported at p < 0.05 using the nonparametric Mann-Whitney U-test.”

The following sentence has been added to the manuscript and highlighted in green:

“TOPA macrocycles had no ability to reduce A549 cell viability”.

  1. Figure S26. Modify the arrow.

Arrows have been modified in Figures 2 and S26, the necessary changes have been made.

  1. Lines 31-32 and Lines 294-296. The authors claim that "The macrocycle conformation was found to have little effect on protein binding strength, while the native secondary structure of HEWL did not change under interaction with the synthesized thiacalix[4]arenes". But the data in Table 2 Figure 4, and Figures S38-39 suggest something very much otherwise. While the authors' explanation in lines 288-290 is confusing. How come samples were not prepared and analyzed in a consistent manner? Experimental details should have been included. Also see comment#5.

We are grateful the Reviewer for this question. Indeed, the presented study has shown little effect of macrocycle conformation on protein binding strength, since the binding constants of TOPA thiacalixarenes with protein were close (logKa = 5.05 – 5.13). The native secondary structure of HEWL did not change under interaction with the synthesized thiacalix[4]arenes. This fact was confirmed by CD spectroscopy. Table 2 and Figure 4 present data on the formation of aggregates between TOPA compounds and HEWL (CTOPA = CHEWL = 10 µM). In turn, the size and morphology of the presented aggregates differed from each other, primarily due to different macrocycle conformations. The formation of particles with the smallest mean hydrodynamic diameter (84 ± 2 nm) and the smallest PDI (0.11) was observed (Fig. S33) only in the case of TOPA-paco / HEWL mixture. There was a significant stabilization of the system in the presence of protein and a decrease in the size of the particles formed (from 298 ± 22 nm to 84 ± 2 nm) compared to the self-associates formed by TOPA-paco (Table 1). The addition of HEWL to the solutions of TOPA-cone and TOPA-alt resulted in particle enlargement (Figs. S32, S34). Moreover, the zeta potential values of the binary systems (Figs. S35-S37) were found to be close (Table 2) regardless of the macrocycle conformation. The isoelectric point of lysozyme is 11.1, which determines its positive charge in solution and interaction with TOPA due to electrostatic interactions. Eight carboxylate fragments are located on the same side of the macrocyclic platform in the case of TOPA-cone. Therefore, its interaction with HEWL may be most effective. The lipophilic tert-butyl fragments are located on the opposite side of the macrocyclic platform and interacted only with the same lipophilic tert-butyl groups of TOPA-cone, resulting in layer-by-layer assembly of TOPA-cone with the protein. This assembly led to both the aggregates enlargement and stability decreasing of such systems. The interaction of TOPA-alt with HEWL led to most dramatic changes in particle diameter as hypothesized above. In turn, six carboxyl moieties interacted with the HEWL molecule and two carboxyl moiety is located on the opposite side of the macrocyclic platform in the case of TOPA-paco. It prevented layer-by-layer assembly compared to TOPA-cone and determines the smallest size of the formed aggregates.

Samples for DLS and TEM were prepared under the same conditions (concentration and solvent). However, DLS makes it possible to estimate the size of particles in solution, while TEM and SEM allow estimating the size of aggregates in the film. The solvation layer of the aggregate is removed when organic samples are studied, resulting in a difference in the aggregates sizes obtained by DLS and TEM/SEM.

You can find all experimental details regarding Sections 2.4 & 2.5 in Supplementary Information, namely Sections 1.5-1.9 in Supplementary section.

Necessary changes were made to the manuscript and have been highlighted in green.

  1. Supporting figures. Figure captions should be place below the corresponding figure.

Necessary changes were made to the Supplementary information.

  1. Too many references (in total 85). Some cited papers are from the same authors, verging on self-citation.

Thank you for your recommendation. This large number of references is explained by necessity to fully disclose the subject of the research and to provide the most significant examples to confirm the hypothesis. Some references are necessary to explain and compare the obtained results with previously published work, including that of the authors of the manuscript. Moreover, the rules of International Journal of Molecular Sciences do not contain any restrictions on the total number of references. It should be noted that two references of authors were deleted and the total number of references decreased in this way to 83.

Round 2

Reviewer 1 Report

Comments and Suggestions for Authors I have no further comments or reservations. The authors of
the manuscript corrected the manuscript according to my suggestions.
I recommend the manuscript for acceptance.

Reviewer 2 Report

Comments and Suggestions for Authors

The authors have taken into account the reviewer's comments and made revisions to the initial manuscript as suggested. Therefore, I would like to recommend the current version for publication.